# OPTIMAL COMPLETION DISTILLATION FOR SEQUENCE LEARNING

**Sara Sabour, William Chan, Mohammad Norouzi**
`{sasabour, williamchan, mnorouzi}@google.com`
Google Brain

## ABSTRACT

We present *Optimal Completion Distillation* (OCD), a training procedure for optimizing sequence to sequence models based on edit distance. OCD is efficient, has no hyper-parameters of its own, and does not require pretraining or joint optimization with conditional log-likelihood. Given a partial sequence generated by the model, we first identify the set of optimal suffixes that minimize the total edit distance, using an efficient dynamic programming algorithm. Then, for each position of the generated sequence, we define a target distribution that puts an equal probability on the first token of each optimal suffix. OCD achieves the state-of-the-art performance on end-to-end speech recognition, on both Wall Street Journal and Librispeech datasets, achieving $9.3\%$ and $4.5\%$ word error rates, respectively.

## 1 INTRODUCTION

Recent advances in natural language processing and speech recognition hinge on the development of expressive neural network architectures for sequence to sequence (seq2seq) learning (Sutskever et al., 2014; Bahdanau et al., 2015). Such encoder-decoder architectures are adopted in both machine translation (Bahdanau et al., 2015; Wu et al., 2016; Hassan et al., 2018) and speech recognition systems (Chan et al., 2016; Bahdanau et al., 2016a; Chiu et al., 2017) achieving impressive performance above traditional multi-stage pipelines (Koehn et al., 2007; Povey et al., 2011). Improving the building blocks of seq2seq models can fundamentally advance machine translation and speech recognition, and positively impact other domains such as image captioning (Xu et al., 2015), parsing (Vinyals et al., 2015), summarization (Rush et al., 2015), and program synthesis (Zhong et al., 2017).

To improve the key components of seq2seq models, one can either design better architectures, or develop better learning algorithms. Recent architectures using convolution (Gehring et al., 2017) and self attention (Vaswani et al., 2017) have proved to be useful, especially to facilitate efficient training. On the other hand, despite many attempts to mitigate the limitations of Maximum Likelihood Estimation (MLE) (Ranzato et al., 2016; Wiseman and Rush, 2016; Norouzi et al., 2016; Bahdanau et al., 2017; Leblond et al., 2018), MLE is still considered the dominant approach for training seq2seq models. Current alternative approaches require pre-training or joint optimization with conditional log-likelihood. They are difficult to implement and require careful tuning of new hyper-parameters (*e.g.* mixing ratios). In addition, alternative approaches typically do not offer a substantial performance improvement over a well tuned MLE baseline, especially when label smoothing (Pereyra et al., 2017; Edunov et al., 2018) and scheduled sampling (Bengio et al., 2015) are used.

In this paper, we borrow ideas from search-based structured prediction (Daumé et al., 2009; Ross et al., 2011) and policy distillation (Rusu et al., 2016) and develop an efficient algorithm for optimizing seq2seq models based on edit distance[1]. Our key observation is that given an arbitrary prefix (*e.g.* a partial sequence generated by sampling from the model), we can *exactly* and *efficiently* identify all of the suffixes that result in a minimum total edit distance (*v.s.* the ground truth target). Our training procedure, called *Optimal Completion Distillation (OCD)*, is summarized as follows:

1. We always train on prefixes generated by sampling from the model that is being optimized.
2. For each generated prefix, we identify all of the optimal suffixes that result in a minimum total edit distance *v.s.* the ground truth target using an efficient dynamic programming algorithm.
3. We teach the model to *optimally extend* each generated prefix by maximizing the average log probability of the first token of each optimal suffix identified in step 2.

---

[1] Edit distance between two sequences $\mathbf{u}$ and $\mathbf{v}$ is the minimum number of insertion, deletion, and substitution edits required to convert $\mathbf{u}$ to $\mathbf{v}$ and *vice versa*.

The proposed OCD algorithm is efficient, straightforward to implement, and has no tunable hyper-parameters of its own. Our key contributions include:

- We propose OCD, a stand-alone algorithm for optimizing seq2seq models based on edit distance. OCD is scalable to real-world datasets with long sequences and large vocabularies, and consistently outperforms Maximum Likelihood Estimation (MLE) by a large margin.
- Given a target sequence of length $m$ and a generated sequence of length $n$, we present an $O(nm)$ algorithm that identifies all of the optimal extensions for each prefix of the generated sequence.
- We demonstrate the effectiveness of OCD on end-to-end speech recognition using attention-based seq2seq models. On the Wall Street Journal dataset, OCD achieves a Character Error Rate (CER) of $3.1\%$ and a Word Error Rate (WER) of $9.3\%$ without language model rescoring, outperforming all prior work (Table 4). On Librispeech, OCD achieves state-of-the-art WER of $4.5\%$ on "test-clean" and $13.3\%$ on "test-other" sets (Table 5).

## 2  BACKGROUND: SEQUENCE LEARNING WITH MLE

Given a dataset of input output pairs $\mathcal{D} \equiv \{(\mathbf{x}, \mathbf{y}^*)_i\}_{i=1}^N$, we are interested in learning a mapping $\mathbf{x} \to \mathbf{y}$ from an input $\mathbf{x}$ to a target output sequence $\mathbf{y}^* \in \mathcal{Y}$. Let $\mathcal{Y}$ denote the set of all sequences of tokens from a finite vocabulary $\mathcal{V}$ with variable but finite lengths. Often learning a mapping $\mathbf{x} \to \mathbf{y}$ is formulated as optimizing the parameters of a conditional distribution $p_\theta(\mathbf{y} \mid \mathbf{x})$. Then, the final sequence prediction under the probabilistic model $p_\theta$ is performed by exact or approximate inference (*e.g.* via beam search) as:

$$\hat{\mathbf{y}} \approx \operatorname{argmax}_{\mathbf{y} \in \mathcal{Y}} p_\theta(\mathbf{y} \mid \mathbf{x}) . \tag{1}$$

Similar to the use of log loss for supervised classification, the standard approach to optimize the parameters $\theta$ of the conditional probabilistic model entails maximizing a conditional log-likelihood objective, $\mathcal{O}_{\mathrm{MLE}}(\theta) = \mathbb{E}_{(\mathbf{x},\mathbf{y}^*) \sim p_\mathcal{D}} \log p_\theta(\mathbf{y}^* \mid \mathbf{x})$. This approach to learning the parameters is called Maximum Likelihood Estimation (MLE) and is commonly used in sequence to sequence learning.

Sutskever et al. (2014) propose the use of recurrent neural networks (RNNs) for *autoregressive* seq2seq modeling to tractably optimize $\mathcal{O}_{\mathrm{MLE}}(\theta)$. An autoregressive model estimates the conditional probability of the target sequence given the source one token at a time, often from left-to-right. A special *end-of-sequence* token is appended at the end of all of target sequences to handle variable length. The conditional probability of $\mathbf{y}^*$ given $\mathbf{x}$ is decomposed via the chain rule as,

$$p_\theta(\mathbf{y}^* \mid \mathbf{x}) \equiv \prod_{t=1}^{|\mathbf{y}^*|} p_{\theta,t}(y_t^* \mid \mathbf{y}_{<t}^*, \mathbf{x}) , \tag{2}$$

where $\mathbf{y}_{<t}^* \equiv (y_1^*, \ldots, y_{t-1}^*)$ denotes a prefix of the sequence $\mathbf{y}^*$. To estimate the probability of a token $a$ given a prefix $\mathbf{y}_{<t}^*$ and an input $\mathbf{x}$, denoted $p_{\theta,t}(a \mid \mathbf{y}_{<t}^*, \mathbf{x})$, different architectures have been proposed. Some papers (*e.g.* Britz et al. (2017)) have investigated the use of LSTM (Hochreiter and Schmidhuber, 1997) and GRU (Cho et al., 2014) cells, while others proposed new architectures based on soft attention (Bahdanau et al., 2015), convolution (Gehring et al., 2017), and self-attention (Vaswani et al., 2017). Nonetheless, all of these techniques rely on MLE for learning,

$$\mathcal{O}_{\mathrm{MLE}}(\theta) = \mathbb{E}_{(\mathbf{x},\mathbf{y}^*) \sim p_\mathcal{D}} \sum_{t=1}^{|\mathbf{y}^*|} \log p_{\theta,t}(y_t^* \mid \mathbf{y}_{<t}^*, \mathbf{x}) , \tag{3}$$

where $p_\mathcal{D}$ denotes the empirical data distribution, uniform across the dataset $\mathcal{D}$. We present a new objective function for optimizing autoregressive seq2seq models applicable to any neural architecture.

### 2.1  LIMITATIONS OF MLE FOR AUTOREGRESSIVE MODELS

In order to maximize the conditional log-likelihood (3) of an autoregressive seq2seq model (2), one provides the model with a *prefix* of $t - 1$ tokens from the ground truth target sequence, denoted $\mathbf{y}_{<t}^*$, and maximizes the log-probability of $y_t^*$ as the next token. This resembles a teacher walking a student through a sequence of perfect decisions, where the student learns as a passive observer. However, during inference one uses beam search (1), wherein the student needs to generate each token $\hat{\mathbf{y}}_t$ by conditioning on its own previous outputs, *i.e.* $\hat{\mathbf{y}}_{<t}$ instead of $\mathbf{y}_{<t}^*$. This creates a discrepancy between training and test known as *exposure bias* (Ranzato et al., 2016). Appendix B expands this further.

Concretely, we highlight two limitations with the use of MLE for autoregressive seq2seq modeling:

1. There is a mismatch between the prefixes seen by the model during training and inference. When the distribution of $\hat{\mathbf{y}}_{<t}$ is different from the distribution of $\mathbf{y}^*_{<t}$, then the student will find themselves in a novel situation that they have not been trained for. This can result in poor generalization, especially when the training set is small or the model size is large.
2. There is a mismatch between the training loss and the task evaluation metric. During training, one optimizes the log-probability of the ground truth output sequence, which is often different from the task evaluation metric (*e.g.* edit distance for speech recognition).

There has been a recent surge of interest in understanding and mitigating the limitations of MLE for autoregressive seq2seq modeling. In Section 4 we discuss prior work in detail after presenting our approach below.

## 3 OPTIMAL COMPLETION DISTILLATION

To alleviate the mismatch between inference and training, we *never* train on ground truth target sequences. Instead, we always train on sequences generated by sampling from the current model that is being optimized. Let $\widetilde{\mathbf{y}}$ denote a sequence generated by sampling from the current model, and $\mathbf{y}^*$ denote the ground truth target. Applying MLE to autoregressive models casts the problem of sequence learning as optimizing a mapping $(\mathbf{x}, \mathbf{y}^*_{<t}) \to y^*_t$ from ground truth prefixes to correct next tokens. By contrast, the key question that arises when training on model samples is the choice of targets for learning a similar mapping $(\mathbf{x}, \widetilde{\mathbf{y}}_{<t}) \to ??$ from generated prefixes to next tokens. Instead of using a set of pre-specified targets, OCD solves a prefix-specific problem to find optimal extensions that lead to the best completions according to the task evaluation metric. Then, OCD encourages the model to extend each prefix with the set of optimal choices for the next token.

Our notion of optimal completion depends on the task evaluation metric denoted $R(\cdot, \cdot)$, which measures the similarity between two complete sequences, *e.g.* the ground truth target *v.s.* a generated sequence. Edit distance is a common task metric. Our goal in sequence learning is to train a model, which achieves high scores of $R(\mathbf{y}^*, \widetilde{\mathbf{y}})$. Drawing connection with the goal of reinforcement learning (Sutton and Barto, 1998), let us recall the notion of optimal Q-values. Optimal Q-values for a state-action pair $(s, a)$, denoted $Q^*(s, a)$, represent the maximum future reward that an agent can accumulate after taking an action $a$ at a state $s$ by following with optimal subsequent actions. Similarly, we define Q-values for a prefix $\widetilde{\mathbf{y}}_{<t}$ and the extending token $a$, as the maximum score attainable by concatenating $[\widetilde{\mathbf{y}}_{<t}, a]$ with an optimal suffix $\mathbf{y}$ to create a full sequence $[\widetilde{\mathbf{y}}_{<t}, a, \mathbf{y}]$. Formally,

$$\forall a \in \mathcal{V}, \qquad Q^*(\widetilde{\mathbf{y}}_{<t}, a) \;=\; \max_{\mathbf{y} \in \mathcal{Y}} R(\mathbf{y}^*, [\widetilde{\mathbf{y}}_{<t}, a, \mathbf{y}]) \,. \tag{4}$$

Then, the optimal extension for a prefix $\widetilde{\mathbf{y}}_{<t}$ can be defined as tokens that attain the maximal Q-values, *i.e.* $\operatorname{argmax}_a Q^*(\widetilde{\mathbf{y}}_{<t}, a)$. This formulation allows for a prefix $\widetilde{\mathbf{y}}_{<t}$ to be sampled on-policy from the model $p_\theta$, or drawn off-policy in any way. Table 1 includes an example ground truth target from the Wall Street Journal dataset and the corresponding generated sample from a model. We illustrate that for some prefixes there exist more than a single optimal extension leading to the same edit distance.

Given $Q$-values for our prefix-token pairs, we use an exponential transform followed by normalization to convert Q-values to a soft optimal policy over the next token extension,

$$\pi^*(a \mid \widetilde{\mathbf{y}}_{<t}) = \frac{\exp(Q^*(\widetilde{\mathbf{y}}_{<t}, a)/\tau)}{\sum_{a'} \exp\left(Q^*(\widetilde{\mathbf{y}}_{<t}, a')/\tau\right)} \,, \tag{5}$$

where $\tau \geq 0$ is a temperature parameter. Note the similarity of $\tau$ and the label smoothing parameter helpful within MLE. In our experiments, we used the limit of $\tau \to 0$ resulting in hard targets and no hyper-parameter tuning.

Given a training example $(\mathbf{x}, \mathbf{y}^*)$, we first draw a full sequence $\widetilde{\mathbf{y}} \sim p_\theta(\cdot \mid \mathbf{x})$ *i.i.d.* from the current model, and then minimize a per-step KL divergence between the optimal policy and the model distribution over the next token extension at each time step $t$. The OCD objective is expressed as,

$$\mathcal{O}_{\text{OCD}}(\theta) \;=\; \mathbb{E}_{(\mathbf{x}, \mathbf{y}^*) \sim p_\mathcal{D}} \mathbb{E}_{\widetilde{\mathbf{y}} \sim p_\theta(\cdot|\mathbf{x})} \sum_{t=1}^{|\widetilde{\mathbf{y}}|} \text{KL}\left(\pi^*(\cdot \mid \widetilde{\mathbf{y}}_{<t}) \,\|\, p_{\theta,t}(\cdot \mid \widetilde{\mathbf{y}}_{<t}, \mathbf{x})\right) \,. \tag{6}$$

For every prefix $\widetilde{\mathbf{y}}_{<t}$, we compute the optimal $Q$-values and use (5) to construct the optimal policy distribution $\pi^*$. Then, we distill the knowledge of the optimal policy for each prefix $\widetilde{\mathbf{y}}_{<t}$ into the parametric model using a KL loss. For the important class of sequence learning problems where *edit*

| | |
|---|---|
| Target sequence $\mathbf{y}^*$ | `a s _ h e _ t a l k s _ h i s _ w i f e` |
| Generated sequence $\widetilde{\mathbf{y}}$ | `a s _ e e _ t a l k s _ w h o s e _ w i f e` |
| Optimal extensions for edit distance (OCD targets) | `a s _ h e _ t a l k s _ h i i s _ _ w i f e`
`        h              h    i    w`
`        _` |

Table 1: A sample sequence $\mathbf{y}^*$ from the Wall Street Journal dataset, where the model's prediction $\widetilde{\mathbf{y}}$ is not perfect. The optimal next characters for each prefix of $\widetilde{\mathbf{y}}$ based on edit distance are shown in blue. For example, for the prefix "as_e" there are 3 optimal next characters of "e", "h", and "_". All of these 3 characters when combined with proper suffixes will result in a total edit distance of 1.

*distance* is the evaluation metric, we develop a dynamic programming algorithm to calculate optimal $Q$-values exactly and efficiently for all prefixes of a sequence $\widetilde{\mathbf{y}}$, discussed below.

### 3.1 OPTIMAL Q-VALUES FOR EDIT DISTANCE

We propose a dynamic programming algorithm to calculate optimal Q-values exactly and efficiently for the reward metric of negative edit distance, *i.e.* $R(\mathbf{y}^*, \widetilde{\mathbf{y}}) = -D_{\text{edit}}(\mathbf{y}^*, \widetilde{\mathbf{y}})$. Given two sequences $\mathbf{y}^*$ and $\widetilde{\mathbf{y}}$, we compute the Q-values for every prefix $\widetilde{\mathbf{y}}_{<t}$ and any extending token $a \in \mathcal{V}$ with an asymptotic complexity of $O(|\mathbf{y}^*|.|\widetilde{\mathbf{y}}| + |\mathcal{V}|.|\widetilde{\mathbf{y}}|)$. Assuming that $|\mathbf{y}^*| \approx |\widetilde{\mathbf{y}}| \leq |\mathcal{V}|$, our algorithm does not increase the time complexity over MLE, since computing the cross-entropy losses in MLE also requires a complexity of $O(|\mathbf{y}^*|.|\mathcal{V}|)$. When this assumption does not hold, *e.g.* genetic applications, OCD is less efficient than MLE. However, in practice, the wall clock time is dominated by the forward and backward passes of a neural networks, and the OCD cost is often negligible. We discuss the efficiency of OCD further in Appendix A.

Recall the Levenshtein algorithm (Levenshtein, 1966) for calculating the minimum number of edits (insertion, deletion and substitution) required to convert sequences $\widetilde{\mathbf{y}}$ and $\mathbf{y}^*$ to each other based on,

$$
\begin{cases}
D_{\text{edit}}(\widetilde{\mathbf{y}}_{<-1}, :) = \infty \\
D_{\text{edit}}(:, \mathbf{y}^*_{<-1}) = \infty \\
D_{\text{edit}}(\widetilde{\mathbf{y}}_{<0}, \mathbf{y}^*_{<0}) = 0 \,,
\end{cases}
\quad
D_{\text{edit}}(\widetilde{\mathbf{y}}_{<i}, \mathbf{y}^*_{<j}) = \min
\begin{cases}
D_{\text{edit}}(\widetilde{\mathbf{y}}_{<i-1}, \mathbf{y}^*_{<j}) + 1 \\
D_{\text{edit}}(\widetilde{\mathbf{y}}_{<i}, \mathbf{y}^*_{<j-1}) + 1 \\
D_{\text{edit}}(\widetilde{\mathbf{y}}_{<i-1}, \mathbf{y}^*_{<j-1}) + \mathbb{1}[\tilde{y}_i \neq y^*_j] \,.
\end{cases}
\tag{7}
$$

Table 2 shows an example edit distance table for sequences "Satrapy" and "Sunday". Our goal is to identify the set of all optimal suffixes $\mathbf{y} \in \mathcal{Y}$ that result in a full sequences $[\widetilde{\mathbf{y}}_{<i}, \mathbf{y}]$ with a minimum edit distance *v.s.* $\mathbf{y}^*$.

**Lemma 1.** *The edit distance resulting from any potential suffix $\mathbf{y} \in \mathcal{Y}$ is lower bounded by $m_i$,*

$$
\forall \mathbf{y} \in \mathcal{Y}, \ D_{edit}([\widetilde{\mathbf{y}}_{<i}, \mathbf{y}], \mathbf{y}^*) \geq \min_{0 \leq j \leq |\mathbf{y}^*|} D_{edit}(\widetilde{\mathbf{y}}_{<i}, \mathbf{y}^*_{<j}) = m_i \,.
\tag{8}
$$

*Proof.* Let's consider the path $P$ that traces $D_{\text{edit}}([\widetilde{\mathbf{y}}_{<i}, \mathbf{y}], \mathbf{y}^*)$ back to $D_{\text{edit}}(\widetilde{\mathbf{y}}_{<0}, \mathbf{y}^*_{<0})$ connecting each cell to an adjacent parent cell, which provides the minimum value among the three options in (7). Such a path for tracing edit distance between "Satrapy" and "Sunday" is shown in Table 2.

Table 2: Each row corresponds to a prefix of "SATRAPY" and shows edit distances with all prefixes of "SUNDAY". We also show OCD targets (optimal extensions) for each prefix, and minimum value along each row, denoted $m_i$ (see (8)). We highlight the trace path for $D_{\text{edit}}$("Satrapy", "Sunday").

| | | Edit Distance Table | | | | | | OCD Targets | $m_i$ |
|---|---|---|---|---|---|---|---|---|---|
| | | S | U | N | D | A | Y | | |
| | **0** | 1 | 2 | 3 | 4 | 5 | 6 | S | **0** |
| S | 1 | **0** | 1 | 2 | 3 | 4 | 5 | U | **0** |
| A | 2 | **1** | **1** | 2 | 3 | 3 | 4 | U, N | **1** |
| T | 3 | 2 | **2** | **2** | 3 | 4 | 4 | U, N, D | **2** |
| R | 4 | **3** | **3** | **3** | **3** | 4 | 5 | U, N, D, A | **3** |
| A | 5 | 4 | 4 | 4 | 4 | **3** | 4 | Y | **3** |
| P | 6 | 5 | 5 | 5 | 5 | **4** | **4** | Y,  | **4** |
| Y | 7 | 6 | 6 | 6 | 6 | 5 | **4** |  | **4** |

Suppose the path $P$ crosses row $i$ at a cell $(i, k)$. Since the operations in (7) are non-decreasing, the edit distance along the path cannot decrease, so $\mathrm{D}_{\mathrm{edit}}([\widetilde{\mathbf{y}}_{<i}, \mathbf{y}], \mathbf{y}^*) \geq \mathrm{D}_{\mathrm{edit}}(\widetilde{\mathbf{y}}_{<i}, \mathbf{y}^*_{<k}) \geq m_i$. □

Then, consider any $k$ such that $\mathrm{D}_{\mathrm{edit}}(\widetilde{\mathbf{y}}_{<i}, \mathbf{y}^*_{<k}) = m_i$. Let $\mathbf{y}^*_{\geq k} \equiv (y^*_k, \dots, y^*_{|\mathbf{y}^*|})$ denote a suffix of $\mathbf{y}^*$. We conclude that $\mathrm{D}_{\mathrm{edit}}([\widetilde{\mathbf{y}}_{<i}, \mathbf{y}^*_{\geq k}], \mathbf{y}^*) = m_i$, because on the one hand there is a particular edit path that results in $m_i$ edits, and on the other hand $m_i$ is a lower bound according to Lemma 1. Hence any such $\mathbf{y}^*_{\geq k}$ is an optimal suffix for $\widetilde{\mathbf{y}}_{<i}$. Further, it is straightforward to prove by contradiction that the set of optimal suffixes is limited to suffixes $\mathbf{y}^*_{\geq k}$ corresponding to $\mathrm{D}_{\mathrm{edit}}(\widetilde{\mathbf{y}}_{<i}, \mathbf{y}^*_{<k}) = m_i$.

Since the set of optimal completions for $\widetilde{\mathbf{y}}_{<i}$ is limited to $\mathbf{y}^*_{\geq k}$, the only extensions that can lead to maximum reward are the starting token of such suffixes ($y^*_k$). Since $\mathrm{D}_{\mathrm{edit}}(\widetilde{\mathbf{y}}_{<i}, \mathbf{y}^*_{<k}) = m_i$ as well, we can identify the optimal extensions by calculating the edit distances between all prefixes of $\widetilde{\mathbf{y}}$ and all prefixes of $\mathbf{y}^*$ which can be efficiently calculated by dynamic programming in $\mathcal{O}(|\widetilde{\mathbf{y}}|.|\mathbf{y}^*|)$. For a prefix $\widetilde{\mathbf{y}}_{<i}$ after we calculate the minimum edit distance $m_i$ among all prefixes of $\mathbf{y}^*$, we set the $Q^*(\widetilde{\mathbf{y}}_{<i}, y^*_k) = -m_i$ for all $k$ where $\mathbf{y}^*_{<k}$ has edit distance equal to $m_i$. We set the $Q^*$ for any other token to $-m_i - 1$. We provide the details of our modified Levenshtein algorithm to efficiently compute the $Q^*(\widetilde{\mathbf{y}}_{<i}, a)$ for all $i$ and $a$ in Appendix A.

## 4 RELATED WORK

Our work builds upon Learning to Search (Daumé III and Marcu, 2005) and Imitation Learning techniques (Ross et al., 2011; Ross and Bagnell, 2014; Sun et al., 2018), where a student policy is optimized to imitate an expert teacher. DAgger (Ross et al., 2011) in particular is closely related, where a dataset of trajectories from an expert teacher is aggregated with samples from past student models, and a policy is optimized to mimic a given expert policy $\pi^*$ at various states. Similarly, OCD aims to mimic an optimal policy $\pi^*$ at all prefixes, but in OCD, the behavior policy is directly obtained from an online student. Further, the oracle policy is not provided during training, and we obtain the optimal policy by finding optimal $Q$-values. AggreVaTeD (Sun et al., 2017) assumes access to an unbiased estimate of Q-values and relies on variance reduction techniques and conjugate gradients to address a policy optimization problem. OCD calculates exact Q-values and uses regular SGD for optimization. Importantly, our *roll-in* prefixes are drawn only from the student model, and we do not require mixing in ground truth (*a.k.a.* expert) samples. Cheng and Boots (2018) showed that mixing in ground truth samples is an essential regularizer for value aggregation convergence in imitation learning.

Our work is closely related to Policy Distillation (Rusu et al., 2016), where a Deep Q-Network (DQN) agent (Mnih et al., 2015) that is previously optimized is used as the expert teacher. Then, action sequences are sampled from the teacher and the learned Q-value estimates are distilled (Hinton et al., 2014) into a smaller student network using a KL loss. OCD adopts a similar loss function, but rather than estimating Q-values using bootstrapping, we estimate exact $Q$-values using dynamic programming. Moreover, we draw samples from the student rather than the teacher.

Similar to OCD, the learning to search (L2S) techniques such as LOLS (Chang et al., 2015) and Goodman et al. (2016) also attempt to estimate the Q-values for each state-action pair. Such techniques examine multiple *roll-outs* of a generated prefix and aggregate the return values. SeaRNN (Leblond et al., 2018) approximates the cost-to-go for each token by computing the task loss for as many roll-outs as the vocabulary size at each time step with a per step complexity of $O(VT)$. It is often difficult to scale approaches based on multiple roll-outs to real world datasets, where either the sequences are long or the vocabulary is large. OCD exploits the special structure in edit distance and find exact Q-values efficiently in $O(V + T)$ per step. Unlike L2S and SeaRNN, which require ground truth prefixes to stabilize training, we solely train on model samples.

Approaches based on Reinforcement Learning (RL) have also been applied to sequence prediction problems, including REINFORCE (Ranzato et al., 2016), Actor-Critic (Bahdanau et al., 2017) and Self-critical Sequence Training (Rennie et al., 2017). These methods sample sequences from the model's distribution and backpropagate a sequence-level task objective (*e.g.* edit distance). Beam Search Optimization (Wiseman and Rush, 2016) and Edit-based Minimum Bayes Risk (EMBR) (Prabhavalkar et al., 2018) is similar, but the sampling procedure is replaced with beam search. These training methods suffer from high variances and credit assignment problems. By contrast, OCD takes advantage of the decomposition of the sequence-level objective into token level optimal completion targets. This reduces the variance of the gradient and stabilizes the model. Crucially, unlike most

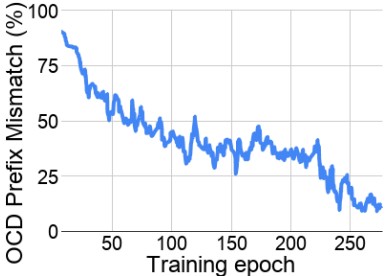

Figure 1: Fraction of OCD training prefix tokens on WSJ which does not match ground truth.

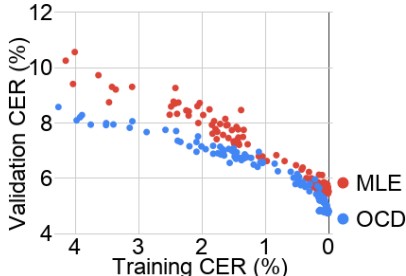

Figure 2: WSJ validation Character Error Rate (CER) per training CER for MLE and OCD.

RL-based approaches, we neither need MLE pretraining or joint optimization with log-likelihood. Bahdanau et al. (2016b) also noticed some of the nice structure of edit distance, but they optimize the model by regressing its outputs to edit distance values leading to suboptimal performance. Rather, we first construct the optimal policy and then use knowledge distillation for training. Independently, Karita et al. (2018) also decomposed edit distance into the contribution of individual tokens and used this decomposition within the EMBR framework. That said, Karita et al. (2018) do not theoretically justify this particular choice of decomposition and report high variance in their gradient estimates.

Reward Augmented Maximum Likelihood (RAML) (Norouzi et al., 2016) and its variants (Ma et al., 2017; Elbayad et al., 2018; Wang et al., 2018) are also similiar to RL-based approaches. Instead of sampling from the model's distribution, RAML samples sequences from the true exponentiated reward distribution. However, sampling from the true distribution is often difficult and intractable. RAML suffers from the same problems as RL-based methods in credit assignment. SPG (Ding and Soricut, 2017) changes the policy gradient formulation to sample from a reward shaped model distribution. Therefore, its samples are closer than RAML to the model's samples. In order to facilitate sampling from their proposed distribution SPG provides a heuristic to decompose ROUGE score. Although SPG has a lower variance due to their biased samples, it suffers from the same problems as RAML and RL-based methods in credit assignment.

Generally, OCD excels at training from scratch, which makes it an ideal substitution for MLE. Hence, OCD is orthogonal to methods which require MLE pretraining or joint optimization.

## 5 EXPERIMENTS

We conduct our experiments on speech recogntion on the Wall Street Journal (WSJ) (Paul and Baker, 1992) and Librispeech (Panayotov et al., 2015) benchmarks. We only compare end-to-end speech recognition approaches that do not incorporate language model rescoring. On both WSJ and Librispeech, our proposed **OCD** (Optimal Completion Distillation) algorithm significantly outperforms our own strong baselines including **MLE** (Maximum Likelihood Estimation with label smoothing) and **SS** (scheduled sampling with a well-tuned schedule). Moreover, OCD significantly outperforms all prior work, achieving a new state-of-the-art on two competitive benchmarks.

### 5.1 WALL STREET JOURNAL

The WSJ dataset is readings of three separate years of the Wall Street Journal. We use the standard configuration of si284 for training, dev93 for validation and report both test Character Error Rate (CER) and Word Error Rate (WER) on eval92. We tokenize the dataset to English characters and punctuation. Our model is an attention-based seq2seq network with a deep convolutional frontend as used in Zhang et al. (2017). During inference, we use beam search with a beam size of 16 for all of our models. We describe the architecture and hyperparameter details in Appendix C. We first analyze some key characteristics of the OCD model separately, and then compare our results with other baselines and state-of-the-art methods.

**Training prefixes and generalization**. We emphasize that during training, the generated prefixes sampled from the model do not match the ground truth sequence, even at the end of training. We

Table 3: WSJ Character Error Rate (CER) and Word Error Rate (WER) of different baselines. Schedule Sampling optimizes for *Hamming* distance and mixes samples from the model and ground truth with a *probability* schedule (start-of-training → end-of-training). OCD *always* samples from the model and optimizes for *all* characters which minimize *Edit* distance. Optimal Completion Target optimizes for *one* character which minimizes edit distance and another criteria (shortest or same #words).

| Training Strategy | CER | WER |
|---|---|---|
| Schedule Sampling ($1.0 \rightarrow 1.0$) | 12.1 | 35.6 |
| Schedule Sampling ($0.0 \rightarrow 1.0$) | 3.8 | 11.7 |
| Schedule Sampling ($0.0 \rightarrow 0.55$) | 3.6 | 10.2 |
| Optimal Completion Target (Shortest) | 3.8 | 12.7 |
| Optimal Completion Target (Same #Words) | 3.3 | 10.2 |
| Optimal Completion Distillation | **3.1** | **9.3** |

Table 4: Character Error Rate (CER) and Word Error Rate (WER) results on the end-to-end speech recognition WSJ task. We report results of our Optimal Completion Distillation (OCD) model, and well-tuned implementations of maximum likelihood estimation (MLE) and Scheduled Sampling (SS).

| Model | CER | WER |
|---|---|---|
| **Prior Work** | | |
| CTC (Graves and Jaitly; 2014) | 9.2 | 30.1 |
| CTC + REINFORCE (Graves and Jaitly; 2014) | 8.4 | 27.3 |
| Gram-CTC (Liu et al.; 2017) | - | 16.7 |
| seq2seq (Bahdanau et al.; 2016a) | 6.4 | 18.6 |
| seq2seq + TLE (Bahdanau et al.; 2016b) | 5.9 | 18.0 |
| seq2seq + LS (Chorowski and Jaitly; 2017) | - | 10.6 |
| seq2seq + CNN (Zhang et al.; 2017) | - | 10.5 |
| seq2seq + LSD (Chan et al.; 2017) | - | 9.6 |
| seq2seq + CTC (Kim et al.; 2017) | 7.4 | - |
| seq2seq + TwinNet (Serdyuk et al.; 2018) | 6.2 | - |
| seq2seq + MLE + REINFORCE (Tjandra et al.; 2018) | 6.1 | - |
| **Our Implementation** | | |
| seq2seq + MLE | 3.6 | 10.6 |
| seq2seq + SS | 3.6 | 10.2 |
| **seq2seq + OCD** | **3.1** | **9.3** |

define OCD prefix *mismatch* as the fraction of OCD training tokens that do not match corresponding ground truth training tokens at each position. Assuming that the generated prefix sequence is perfectly matched with the ground truth sequence, then the OCD targets would simply be the following tokens of the ground truth sequence. Hence, OCD becomes equivalent to MLE. Figure 1 shows that OCD prefixes mismatch is more than $25\%$ for the most of the training. This suggests that OCD and MLE are training on very different input prefix trajectories. Further, Figure 2 depicts validation CER as a function of training CER for different model checkpoints during training, where we use beam search on both training and validation sets to obtain CER values. Even at the same training CER, we observe better validation error for OCD, which suggests that OCD improves generalization of MLE, possibly because OCD alleviates the mismatch between training and inference.

**Impact of edit distance.** We further investigate the role of the optimizer by experimenting with different losses. Table 3 compares the test CER and WER of the schedule sampling with a fixed probability schedule of ($1.0 \rightarrow 1.0$) and OCD model. Both of the models are trained only on sampled trajectories. The main difference is their optimizers, where the SS($1.0 \rightarrow 1.0$) model is optimizing the log likelihood of ground truth (*a.k.a.* Hamming distance). The significant drop in CER of SS($1.0 \rightarrow 1.0$) emphasizes the necessity of pretraining or joint training with MLE for models such as SS. OCD is trained from random initialization and does not require MLE pretraining, nor

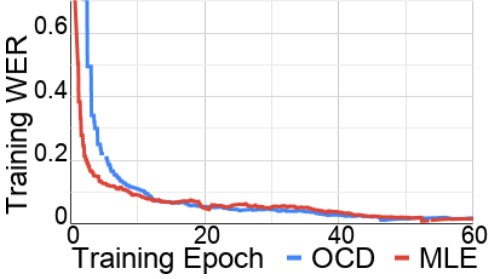 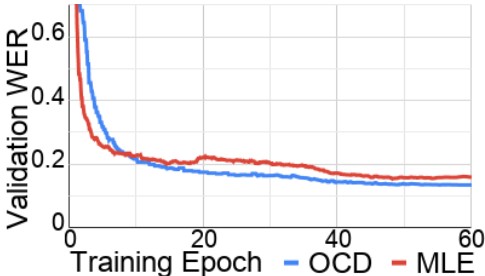

Figure 3: Librispeech training and validation WER per training epoch for OCD and MLE.

does it require joint optimization with MLE. We also emphasize that unlike SS, we do not need to tune an exploration schedule, OCD prefixes are simply always sampled from the model from the start of training. We note that even fine tuning a pre-trained SS model which achieves 3.6% CER with 100% sampling increases the CER to 3.8%. This emphasizes the importance of making the loss a function of the model input prefixes, as opposed to the ground truth prefixes. Appendix D covers another aspect of optimizing Edit distance rather than Hamming distance.

**Target distribution.** Another baseline which is closer to MLE framework is selecting only one correct target. Table 3 compares OCD with several Optimal Completion Target (OCT) models. In OCT, we optimize the log-likelihood of one target, which at each step we pick dynamically based on the minimum edit distance completion similar to OCD. We experiment with several different strategies when there is more than one character that can lead to minimum CER. In the OCT (Shortest), we select the token that would minimize the CER and the final length of the sequence. In the OCT (Same #Words), we select the token that in addition to minimum CER, would lead to the closest number of words to the target sequence. We show that OCD achieves significantly better CER and WER over the other optimization strategies compared in Table 3. This highlights the importance of optimizing for the entire set of optimal completion targets, as opposed to a single target.

**State-of-the-art.** Our model trained with OCD optimizes for CER; we achieve 3.1% CER and 9.3% WER, substantially outperforming our baseline by 14% relatively on CER and 12% relatively on WER. In terms of CER, our work substantially outperforms prior work as compared in Table 4, with the closest being Tjandra et al. (2018) trained with policy gradients on CER. In terms of WER, our work is also outperforming Chan et al. (2017), which uses subword units while our model emits characters.

## 5.2 LIBRISPEECH

For the Librispeech dataset, we train on the full training set (960h audio data) and validate our results on the dev-other set. We report the results both on the "clean" and "other" test set. We use Byte Pair Encoding (BPE) (Sennrich et al., 2016) for the output token segmentation. BPE token set is an open

Table 5: Character Error Rate (CER) and Word Error Rate (WER) on LibriSpeech test sets.

| Model | test-clean | | test-other | |
|---|---|---|---|---|
| | CER | WER | CER | WER |
| Prior Work | | | | |
|     Wav2letter (Collobert et al., 2016) | 6.9 | 7.2 | - | - |
|     Gated ConvNet (Liptchinsky et al., 2017) | - | 6.7 | - | 20.8 |
|     Cold Fusion (Sriram et al., 2018) | 3.9 | 7.5 | 9.3 | 17.0 |
|     Invariant Representation Learning (Liang et al., 2018) | 3.3 | - | 11.0 | - |
|     Pretraining+seq2seq+CTC (Zeyer et al., 2018) | - | 4.9 | - | 15.4 |
| Our Implementation | | | | |
|     seq2seq + MLE | 2.9 | 5.7 | 8.4 | 15.4 |
|     **seq2seq + OCD** | **1.7** | **4.5** | **6.4** | **13.3** |

vocabulary set since it includes the characters as well as common words and n-grams. We use 10k BPE tokens and report both CER and WER as the evaluation metric. We describe the architecture and hyperparameter details in Appendix C.

Fig. 3 shows the validation and training WER curves for MLE and OCD. OCD starts outperforming MLE on training decodings after training for 13 epochs and on validation decodings after 9 epochs. Our MLE baseline achieves $5.7\%$ WER, while OCD achieves $4.5\%$ WER on test-clean ($21\%$ improvement) and improves the state-of-the-art results over Zeyer et al. (2018). test-other is the more challenging test split ranked by the WER of a model trained on WSJ (Panayotov et al., 2015) mainly because readers accents deviate more from US-English accents. On test-other our MLE baseline achieves 15.4%, while our OCD model achieves 13.3% WER, outperforming the 15.4% WER of Zeyer et al. (2018). Table 5 compares our results with other recent works and the MLE baseline on Librispeech.

## 6   CONCLUSION

This paper presents Optimal Completion Distillation (OCD), a training procedure for optimizing autoregressive sequence models base on edit distance. OCD is applicable to on-policy or off-policy trajectories, and in this paper, we demonstrate its effectiveness on samples drawn from the model in an online fashion. Given any prefix, OCD creates an optimal extension policy by computing the exact optimal Q-values via dynamic programming. The optimal extension policy is distilled by minimizing a KL divergence between the optimal policy and the model. OCD does not require MLE initialization or joint optimization with conditional log-likelihood. OCD achieves 3.1% CER and 9.3% WER on the competitive WSJ speech recognition task, and $4.5\%$ WER on Librispeech without any language model. OCD outperforms all published work on end-to-end speech recognition, including our own well-tuned MLE and scheduled sampling baselines without introducing new hyper-parameters.

## ACKNOWLEDGEMENTS

We thank Geoffrey Hinton for his valuable feedback and reviews. We thank Samy Bengio, Navdeep Jaitly, and Jamie Kiros for their help in reviewing the manuscript as well. We thank Zhifeng Chen and Yonghui Wu for their generous technical help.

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

---

1: **for** $j$ in $(0..t)$ **do**
2:     $d_j \leftarrow j + 1$
3: **for** $i$ in $(1..t)$ **do**
4:     minDist $\leftarrow i$
5:     subCost $\leftarrow i - 1$
6:     insCost $\leftarrow i + 1$
7:     **for** $j$ in $(0..t-1)$ **do**
8:         **if** $h_{i-1} = r_j$ **then**
9:             repCost $\leftarrow 0$
10:        **else**
11:            repCost $\leftarrow 1$
12:        cheapest $\leftarrow min(\text{subCost} + \text{repCost}, d_j + 1, \text{insCost})$
13:        subCost $\leftarrow d_j$
14:        insCost $\leftarrow$ cheapest + 1
15:        $d_j \leftarrow$ cheapest
16:        **if** $d_j <$ minDist **then**
17:            minDist $\leftarrow d_j$
18:    **if** minDist $= i$ **then**
19:        $Q_{i,r_1} \leftarrow 1$
20:    **for** $j$ in $(1..t)$ **do**
21:        **if** $d_j =$ minDist **then**
22:            $Q_{i,r_{j+1}} \leftarrow 1$
23:    **for** all tokens k **do**
24:        $Q_{i,k} \leftarrow Q_{i,k} - 1 -$ minDist
    **return** $Q$

---

## APPENDIX A    OCD ALGORITHM

**Complexity.** The total time complexity for calculating the sequence loss using OCD is $O(T^2 + |V|T)$ where $V$ is the vocabulary size and $T$ is the sequence length. MLE loss has a time complexity of $O(|V|T)$ for calculating the softmax loss at each step. Therefore, assuming that $O(T) \leq O(|V|)$ OCD does not change the time complexity compared to the baseline seq2seq+MLE. The memory cost of the OCD algorithm is $O(T + |V|T) = O(|V|T)$, $O(T)$ for the dynamic programming in line 4 - line 13 of Proc. 1 and $O(|V|T)$ for storing the stepwise $Q$ values. MLE also stores the one-hot encoding of targets at each step with a cost of $O(|V|T)$. Therefore, the memory complexity does not change compared to the MLE baseline either.

Although the loss calculation has the same complexity as MLE, online sampling from the model to generate the input of next RNN cell (as in OCD and SS) is generally slower than reading the ground truth (as in MLE). Therefore, overall a naive implementation of OCD is $\leq 20\%$ slower than our baseline MLE in terms of number of step time. However, since OCD is stand alone and can be trained off-policy, we can also train on stale samples and untie the input generation worker from the training workers. In this case it is as fast as the MLE baseline.

**Run through.** As an example of how this algorithm works, consider the sequence "SUNDAY" as reference and "SATURDAY" as hypothesis. Table A.1 first shows how to extract optimal targets and their respective $Q^*$-values from the table of edit distances between all prefixes of reference and all prefixes of hypothesis. At each row highlighted cells indicate the prefixes which has minimum edit distance in the row. The next character at these indices are the Optimal targets for that row. At each step the $Q^*$-value for the optimal targets is negative of the minimum edit distance and for the non-optimal characters it is one smaller.

Table A.1 also illustrates how appending the optimal completions for the prefix "SA" of the hypothesis can lead to the minimum total edit distance. Concatenating with both reference suffixes, "UNDAY"

and "NDAY" will result in an edit distance of 1. Therefore, predicting "U" or "N" at step 2 can lead to the maximum attainable reward of $(-1)$.

Table A.1: Top: Each row corresponds to a prefix of "SATURDAY" and shows edit distances with all prefixes of "SUNDAY", along with the optimal targets and their $Q^*$-value at that step. The highlighted cells indicate cells with minimum edit distance at each row. Bottom: An example of appending suffixes of "SUNDAY" with minimum edit distance to the prefix "SA".

| | | Edit Distance | | | | | | OCD Targets | Q-values |
|---|---|---|---|---|---|---|---|---|---|
| | | S | U | N | D | A | Y | | |
| | 0 | 1 | 2 | 3 | 4 | 5 | 6 | S | 0 |
| S | 1 | 0 | 1 | 2 | 3 | 4 | 5 | U | 0 |
| A | 2 | 1 | 1 | 2 | 3 | 3 | 4 | U, N | -1 |
| T | 3 | 2 | 2 | 2 | 3 | 4 | 4 | U, N, D | -2 |
| U | 4 | 3 | 2 | 3 | 3 | 4 | 5 | N | -2 |
| R | 5 | 4 | 3 | 3 | 4 | 4 | 5 | N, D | -3 |
| D | 6 | 5 | 4 | 4 | 3 | 4 | 5 | A | -3 |
| A | 7 | 6 | 5 | 5 | 4 | 3 | 4 | Y | -3 |
| Y | 8 | 7 | 6 | 6 | 5 | 4 | 3 |  | -3 |

| | | S | U | N | D | A | Y |
|---|---|---|---|---|---|---|---|
| | 0 | 1 | 2 | 3 | 4 | 5 | 6 |
| S | 1 | 0 | 1 | 2 | 3 | 4 | 5 |
| A | 2 | 1 | 1 | 2 | 3 | 3 | 4 |
| U | | | 1 | | | | |
| N | | | | 1 | | | |
| D | | | | | 1 | | |
| A | | | | | | 1 | |
| Y | | | | | | | 1 |

| | | S | U | N | D | A | Y |
|---|---|---|---|---|---|---|---|
| | 0 | 1 | 2 | 3 | 4 | 5 | 6 |
| S | 1 | 0 | 1 | 2 | 3 | 4 | 5 |
| A | 2 | 1 | 1 | 2 | 3 | 3 | 4 |
| N | | | | 1 | | | |
| D | | | | | 1 | | |
| A | | | | | | 1 | |
| Y | | | | | | | 1 |

## APPENDIX B    EXPOSURE BIAS

A key limitation of teacher forcing for sequence learning stems from the discrepancy between the training and test objectives. One trains the model using conditional log-likelihood $\mathcal{O}_{\text{CLL}}$, but evaluates the quality of the model using empirical reward $\mathcal{O}_{\text{ER}}$.

Unlike teacher forcing and Scheduled Sampling (SS), policy gradient approaches (*e.g.* Ranzato et al. (2016); Bahdanau et al. (2017)) and OCD aim to optimize the empirical reward objective (4) on the training set. We illustrate four different training strategies of MLE, SS, Policy Gradient and OCD in Figure B.1. The drawback of policy gradient techniques is twofold: 1) they cannot easily incorporate ground truth sequence information except through the reward function, and 2) they have difficulty reducing the variance of the gradients to perform proper credit assignment. Accordingly, most policy gradient approaches Ranzato et al. (2016); Bahdanau et al. (2017); Wu et al. (2016) pre-train the model using teacher forcing. By contrast, the OCD method proposed in this paper defines an optimal completion policy $\pi_t^*$ for any *off-policy* prefix by incorporating the ground truth information. Then, OCD optimizes a token level log-loss and alleviates the credit assignment problem. Finally, training is much more stable, and we do not require initialization nor joint optimization with MLE.

There is an intuitive notion of *exposure bias* Ranzato et al. (2016) discussed in the literature as a limitation of teacher forcing. We formalize this notion as follows. One can think of the optimization of the log loss (3) in an autoregressive models as a classification problem, where the input to the classifier is a tuple $(\mathbf{s}, \mathbf{y}_{<t}^*)$ and the correct output is $y_i^*$, where $\mathbf{y}_{<t}^* \equiv (y_1^*, \ldots, y_{t-1}^*)$. Then the training dataset comprises different examples and different prefixes of the ground truth sequence. The key challenge is that once the model is trained, one should not expect the model to generalize to a new prefix $\mathbf{y}_{<t}$ that does not come from the training distribution of $P(\mathbf{y}_{<t}^*)$. This problem can become severe as $\mathbf{y}_{<t}$ becomes more dissimilar to correct prefixes. During inference, when one conducts beam search with a large beam size then one is more likely to discover wrong generalization of $p_\theta(\hat{y}_t|\hat{\mathbf{y}}_{<t}, \mathbf{x})$, because the sequence is optimized globally. A natural strategy to remedy this issue

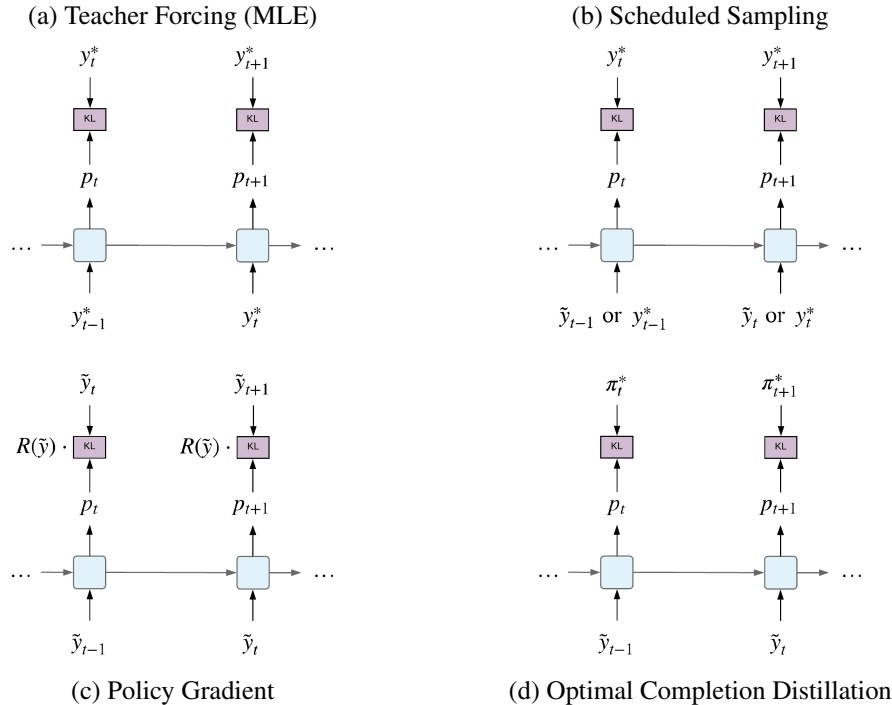

Figure B.1: Illustration of different training strategies for autoregressive sequence models. (a) Teacher Forcing: the model conditions on correct prefixes and is taught to predict the next ground truth token. (b) Scheduled Sampling: the model conditions on tokens either from ground truth or drawn from the model and is taught to predict the next ground truth token regardless. (c) Policy Gradient: the model conditions on prefixes drawn from the model and is encouraged to reinforce sequences with a large sequence reward $R(\tilde{y})$. (d) Optimal Completion Distillation: the model conditions on prefixes drawn from the model and is taught to predict an optimal completion policy $\pi^*$ specific to the prefix.

is to train on arbitrary prefixes. Unlike the aforementioned techniques OCD can train on any prefix given its *off-policy* nature.

Figure B.2 illustrates how increasing the beam size for MLE and SS during inference decreases their performance on WSJ datasets to above $11\%$ WER. OCD suffers a degradation in the performance too but it never gets above $10\%$ WER.

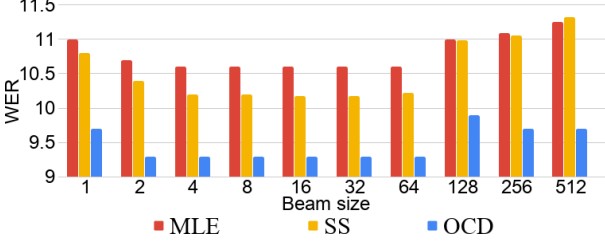

Figure B.2: Word Error Rate (WER) of WSJ with MLE, SS and OCD for different beam sizes.

## APPENDIX C  ARCHITECTURE

**WSJ.** The input audio signal is converted into 80-dimensional filterbank features computed every 10ms with delta and delta-delta acceleration, normalized with per-speaker mean and variance generated by Kaldi (Povey et al., 2011). Our encoder uses 2-layers of convolutions with $3 \times 3$ filters, stride $2 \times 2$ and 32 channels, followed by a convolutional LSTM with 1D-convolution of filter width 3,

followed by 3 LSTM layers with 256 cell size. We also apply batch-normalization between each layer in the encoder. The attention-based decoder is a 1-layer LSTM with 256 cell size with content-based attention. We use Xavier initializer (Glorot and Bengio, 2010) and train our models for 300 epochs of batch size 8 with 8 async workers. We separately tune the learning rate for our baseline and OCD model, 0.0007 for OCD vs 0.001 for baseline. We apply a single 0.01 drop of learning rate when validation CER plateaus, the same as for our baseline. Both happen around 225 epoch. We implemented our experiments[2] in TensorFlow (Abadi et al., 2016).

**Librispeech.** Since the dataset is larger than WSJ, we use a larger batch size of 16, smaller learning rate of 0.0005 for baseline and 0.0003 for OCD. Models are trained for 70 epochs. We remove the convolutional LSTM layers of the encoder, increase the number of LSTM layers in the encoder to 6, and increase the LSTM cell size to 384. All other configs are the same as the WSJ setup.

## APPENDIX D   HAMMING DISTANCE VS EDIT DISTANCE DURING TRAINING

Figure D.3 plots the edit distance on training data of OCD and MLE for fixed hamming distances during training. The plot shows that for a fixed Hamming distance (which is the metric that MLE correlates with more), OCD achieves a lower edit distance compared to MLE. This gives evidence that OCD is indeed optimizing for edit distance as intended.

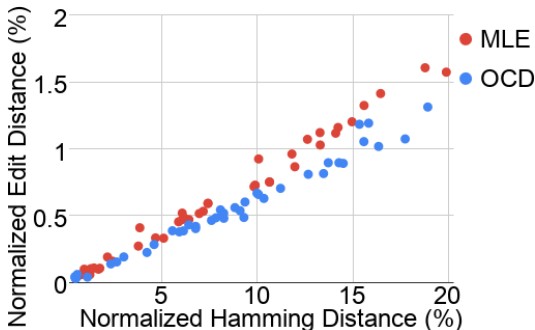

Figure D.3: WSJ training Character Error Rate (CER) of MLE and OCD over Character Accuracy at different checkpoints during training.

---

[2]We are in the process of releasing the code for OCD.

