# OpenReview forum: "Optimal Completion Distillation for Sequence Learning"
_ICLR.cc/2019/Conference_

### Official Review · AnonReviewer1 · 2018-10-30
**A nice twist to seq2seq models**

**Rating:** 6
**Confidence:** 3

**Review:**

The paper considers a shortcoming of sequence to sequence models trained using maximum likelihood estimation. In particular, a model trained in this way can be biased in the sense that training sequences typically have different sets of prefixes compared to test sequences. As a result, at the prediction time the model does not generalize well and for a given input sequence the decoder constructs a label sequence which reflects the training label sequences rather than the actual target label.

To address this shortcoming, the authors propose an approach based on edit distances and the implicit use of given label sequences during training. The main idea is to generate a label sequence with respect to the current parameter vector of a conditional probabilistic model (see Eqs. 2 & 3, as well as the objective in Eq. 6) and then based on the edit distance find the best possible completions for any prefix of that model-based label sequence. The training objective is then defined in Eq. (6): to each element in the output sequence the objective assigns the KL-divergence between a conditional distribution of the next element in the label sequence given a label prefix generated using the current model and the exponential family model based on edit distances given by the prefixes and optimal completions after the position of interest in the label sequence. The objective and the corresponding gradient can be computed efficiently using dynamic programming.

Intuitively, the approach tries to find a parameter vector such that the decoder at a particular instance is likely to construct a label sequence with a small edit distance to the target label. As the training objective now considers all possible elements of the vocabulary given a prefix sequence, it is reasonable to expect that it performs better than MLE which only considers target vocabulary elements given target prefix sequences (e.g., compare Eqs. 3 & 6).

The experiments were conducted on the `Wall Street Journal' and `Librispeech' datasets and the reported results are a significant improvement over the state-of-the-art. I am not an expert in the field and cannot judge the related work objectively but can say that the context for their approach is set appropriately. I would, however, prefer more clarity in the presentation of the approach. This especially applies to the presentation of the approach around Eq. (6). It might not be straightforward for a reader to figure out how the tilde-sequences are obtained. As the objective is non-convex, in order to be able to reproduce the results it would be useful to provide some heuristics for choosing the initial solutions for the parameter vector. In Section 3, please also provide a reference to the appendix so that a reader can understand the conditional probabilistic model.

---

> ### Author Response · Authors · 2018-11-24
> **Thank you for the valuable feedback and review. We have updated the paper accordingly. Our responses follow.**
>
> Dear Reviewer 1,
> Thank you for the valuable feedback and review. We have updated the paper accordingly. Our responses follow.
>
> > “This especially applies to the presentation of the approach around Eq. (6). It might not be straightforward for a reader to figure out how the tilde-sequences are obtained”
>
> We have improved the exposition around Eq. (6). Further, we have emphasised that the tilde-sequences are sampled i.i.d. from the output distribution corresponding to the model that is being optimised.
>
> > “As the objective is non-convex, in order to be able to reproduce the results it would be useful to provide some heuristics for choosing the initial solutions for the parameter vector.”
>
> The initial parameter vector (i.e., the weights of the neural network) are initialized via Xavier initialization [1]. Each weight is randomly drawn from a uniform distribution on the interval [-A, A], where A=sqrt(6. / (fan_in + fan_out)). We included the exact details (and hyper-parameters) in Appendix C. We have confirmed that running the optimization several times from different initializations results in similar results.
>
> [1]: Xavier Glorot and Yoshua Bengio (2010): Understanding the difficulty of training deep feedforward neural networks. International conference on artificial intelligence and statistics.

---

### Official Review · AnonReviewer3 · 2018-11-02
**Exciting approach**

**Rating:** 7
**Confidence:** 4

**Review:**

The authors propose an alternative approach to training seq2seq models, which addresses concerns about exposure bias and about the typical MLE objective being different from the final evaluation metric. In particular, the authors propose to use a dynamic program to compute the optimal continuations of predicted prefixes (during training) in terms of edit distance to the true output, and then use a per-token cross entropy loss, with a target distribution that is uniform over all optimal next-tokens. The authors conduct a number of experiments, and show that this scheme allows them to attain state-of-the-art performance on end-to-end speech recognition, and that they can moreover do this without needing to pretrain the model with the MLE objective.

This is a very nice paper; it is generally well written, it gets excellent results, and it contains a comprehensive set of experiments and analysis. It is also quite exciting to see an approach to alleviating exposure bias that does not require pretraining with MLE. Accordingly, my suggestions mainly relate to the presentation and related work:

 - It seems a bit strange to argue that the proposed approach doesn't increase the time complexity over MLE. While technically true (as the authors note) if the vocabulary is bigger than the sequence lengths, the difference in (on policy) training time will presumably be felt when dealing with very long sequences, or with cases where the number of labels per time-step is small, like in character-level generation or in seq2seq style sequence labeling.

 - I think it's difficult to argue that the proposed approach isn't essentially a modification of imitation learning/learning-to-search algorithms like, say, AggreVaTe or LOLS (Chang et al., ICML 2015). As far as I can tell, the only differences are that cross entropy is used rather than a cost-sensitive classifier, and, perhaps, that the training is done in minibatches (with no aggregation).

 - Relatedly, while it is interesting that the loss uses all the optimal completion tokens, it should be noted that there is much work in transition-based parsing that adopts a learning-to-search approach and uses losses that incorporate multiple optimal next-predictions as given by a "dynamic oracle"; see Goldberg and Nivre (COLING, 2012) and others.

 - I think it's also worth noting that training approaches like MIXER and others can target arbitrary rewards (and not just those where we can efficiently compute optimal next-steps), and so the proposed approach is a compelling competitor to MIXER-like approaches on problems such as machine translation or image captioning only to the extent that training with edit-distance is useful for such problems. Do you have a sense of whether training with edit-distance does indeed improve performance on such tasks?

Pros:
- well written and interesting
- good experiments, results, and analysis

Cons:
 - perhaps slightly more similar to previous work than is argued


Update after author response: thanks for your response; I think the revised paper largely addresses my comments and those of the other reviewers, and I continue to hope it is accepted. Here are two small notes on the related work section of the revised paper:
- In distinguishing OCD from DAgger, you note that the optimal policy is computed rather than provided at training time. In fact, structured prediction applications of SEARN (Daume III et al., 2009, which should also be cited) and DAgger often have this flavor too, such as when using them for sequence labeling (where optimal continuations are calculated based on Hamming distance).
- Please include a reference to Goldberg and Nivre's (2012) dynamic oracle work.

---

> ### Author Response · Authors · 2018-11-24
> **Thank you for the valuable feedback and review. We have updated the paper accordingly. Our responses follow.**
>
> Dear Reviewer 3,
> Thank you for the valuable feedback and review. We have updated the paper accordingly. Our responses follow.
>
> > “It seems a bit strange to argue that the proposed approach doesn't increase the time complexity over MLE. While technically true (as the authors note) if the vocabulary is bigger than the sequence lengths, the difference in (on policy) training time will presumably be felt when dealing with very long sequences …”
>
> Theoretically, OCD and MLE have the same asymptotic time complexity, assuming vocabulary size is larger than the sequence length. When the vocabulary is small, but the sequences are long (e.g. genetic applications), OCD is less efficient than MLE. However, in terms of wall clock time, the OCD cost is amortized by the much larger neural net fprop/bprop computation time. Practically, sampling from the model introduces additional latency given that we will have to wait for previous tokens to be sampled. We have updated paper to state this more clearly and reference the Appendix, where we have provided a more detailed discussion of efficiency.
>
> > “I think it's difficult to argue that the proposed approach isn't essentially a modification of imitation learning/learning-to-search algorithms like, say, AggreVaTe or LOLS (Chang et al., ICML 2015). As far as I can tell, the only differences are that cross entropy is used rather than a cost-sensitive classifier, and, perhaps, that the training is done in minibatches (with no aggregation)”
>
> We have updated the paper, especially the related work section, to make it more clear that OCD builds upon the imitation learning and learning to search algorithms. The use of mini-batches and cross entropy loss rather than cost sensitive classification are two important differences. As opposed to most imitation learning algorithms, we do not require pre-training or mixing with the expert policy. This removes a pesky tunable scheduling hyper-parameter. Another important difference is that most learning to search approaches (e.g. LOLS) resort to roll outs to approximate Q-values, and most imitation learning algorithms (e.g. AggreVaTe) assume availability of oracle completions. By contrast, we propose a dynamic programming algorithm to exactly and efficiently calculate optimal Q-values for edit distance. OCD demonstrates the applicability of imitation learning ideas to modern large-scale sequence-to-sequence problems and achieve state-of-the-art performance on speech recognition.
>
> > “... it should be noted that there is much work in transition-based parsing that adopts a learning-to-search approach and uses losses that incorporate multiple optimal next-predictions as given by a "dynamic oracle"; see Goldberg and Nivre (COLING, 2012) and others”
>
> Thank you for the reference to related work on parsing. We have included the citation. Indeed, Goldberg & Nivre address a structure prediction problem based on a dynamic oracle, which is similar to our notion of optimal completions. That being said, they optimize a graph similarity metric using a margin-based loss, whereas OCD optimizes edit distance based on a KL loss. OCD is immediately applicable to parsing since Goldberg & Nivre can calculate π* for the corresponding graph similarity metric. This is an interesting direction for future work.
>
>  > “I think it's also worth noting that training approaches like MIXER and others can target arbitrary rewards (and not just those where we can efficiently compute optimal next-steps), and so the proposed approach is a compelling competitor to MIXER-like approaches on problems such as machine translation or image captioning only to the extent that training with edit-distance is useful for such problems.”
>
> We acknowledge that policy gradient formulations are more general. However, we believe that OCD is a viable replacement for MLE with better results, given its efficiency and ability to train from scratch. Hence, as of now for optimization of generic reward functions, our suggestion is to replace the MLE pretraining or mixing, with OCD pretraining or mixing. Our preliminary results suggest that EMBR training on top of OCD is more stable and faster than EMBR training on top of MLE.

---

### Official Review · AnonReviewer2 · 2018-11-05
**Interesting learning algorithms for autoregressive models without MLE pretraining**

**Rating:** 7
**Confidence:** 4

**Review:**

Quality and Clarity:
The writing is good and easy to read, and the idea is clearly demonstrated.

Originality:
The idea of never training over the ground-truth sequence, but training on sampled prefix and an optimized suffix is very novel. The similar idea is also related to imitation learning in other domains such as self-driving car where an oracle can give optimal instruction when exploring a new state.

Comments:
The paper proposed a very interesting training algorithm for auto-regressive models especially it does not require any MLE pre-training and can directly optimize from the sampling.

Here are some questions:
(1) The idea should also apply on many “incremental rewards”, for instance, BLEU scores in machine translation, etc. Do you have any comparison? What if the best suffix cannot be found using dynamic programming (when the evaluation metric is not edit-distance, but a sentence-level reward)?
(2) Can the proposed algorithm be applied in other “bigger” tasks such as neural machine translation?
(3) Eq. (6) is not very clear. Do you first sample the whole sequence, and then supervise one token for each step? Or the whole suffix?
(4) Do you have a comparison with the learning efficiency between MLE and OCD? Will it get unstable in the beginning of training as all the samples are wrong.

----------------------------
Missing Reference:
Ding, Nan, and Radu Soricut. "Cold-Start Reinforcement Learning with Softmax Policy Gradient." Advances in Neural Information Processing Systems. 2017.

This paper used a very similar idea as the proposed learning method which relies on incremental rewards to find the “optimal” suffix (for instance, edit-distance is a special example). It would be better to have some discussion,

---

> ### Author Response · Authors · 2018-11-24
> **Thank you for the valuable feedback and review. We have updated the paper accordingly. Our responses follow.**
>
> Dear Reviewer 2,
> Thank you for the valuable feedback and review. We have updated the paper accordingly. Our responses follow.
>
> > “The idea should also apply on many “incremental rewards”, for instance, BLEU scores in machine translation, etc. Do you have any comparison?”
>
> Yes, the main idea is applicable to other structured prediction problems as well. That being said, this paper focuses on edit distance for which exact computation of optimal Q-values is tractable. We are planning to extend OCD to other loss functions in the future.
>
> > “What if the best suffix cannot be found using dynamic programming (when the evaluation metric is not edit-distance, but a sentence-level reward)?”
>
> If the optimal Q-values and optimal suffixes cannot be estimated using dynamic programming, then one can resort to bootstrapping based on Bellman optimality equations (e.g., Q-learning) to approximate optimal Q-values. Further, one can adopt ideas from OCD to consider concatenation of each prefix with various ground truth suffixes to lower bound optimal Q-values and speed up convergence.
>
> > “Can the proposed algorithm be applied in other “bigger” tasks such as neural machine translation?”
>
> OCD is applicable to large-scale datasets because the asymptotic complexity of OCD is the same as MLE. The Librispeech dataset comprises 1,000 hours of speech, we use 10000 tokens and is one of the largest publicly available speech recognition datasets. One can directly apply OCD with edit distance to machine translation, even though the evaluation metric is often BLEU score.
> > “Eq. (6) is not very clear. Do you first sample the whole sequence, and then supervise one token for each step? Or the whole suffix?”
>
> Thanks for the comment. We have improved the exposition around Eq. (6). Yes, we sample the whole sequence, and then each prefix receives the optimal target supervision in the form of one token extension. We did not rely on whole suffix supervision, as it is inefficient and requires some form of MLE again.
>
> > “Do you have a comparison with the learning efficiency between MLE and OCD? Will it get unstable in the beginning of training as all the samples are wrong.”
>
> We added Figure 3 which shows the training and validation WER per training epoch. It shows that OCD is stable in the beginning of training (even if all of the sampled suffixes are wrong), while having a slightly slower curve. We emphasize that while the prefixes deviate from the ground truth initially, the target distribution is always optimal and just a re-alignment of ground truth sequence; consequently OCD can be trained from scratch without any need for MLE warm-starting.
>
> > “Missing Reference: Ding & Soricut … This paper used a very similar idea as the proposed learning method which relies on incremental rewards to find the “optimal” suffix ...“
>
> Thanks for the reference. We acknowledge that previous work has discussed a temporal decomposition of terminal rewards based on the difference of optimal Q-values for consecutive state-action pairs (e.g., see Aggrevate [Ross & Bagnell, 2014]). We did not claim any credit for this. Ding et al. mention reward decomposition for the special case of ROUG score, but we cannot see a proper decomposition of rewards based on optimal Q-value differences in their paper. In our paper, we discuss the special case of edit distance, for which calculation of optimal Q-value differences is tractable. We updated the related work section and included a citation to Ding & Soricut.

---

### Public Comment · (anonymous) · 2018-10-29
**Word-level results**

Hi,

First of all, I'm a big fan of you paper. I thing it reads well and the idea is conceptually simple, yet powerful. I'm especially interested in applying OCD for natural language generation, and was wondering if you had experimented with word-level Seq2Seq models. There is evidence in the litterature [1] that exposure bias is more problematic in character-level vs word-level models, and I'm curious to see if OCD can lead to improvements for word-level models.

Thanks in advance!
-Lucas

[1] https://arxiv.org/abs/1610.09038

---

> ### Author Response · Authors · 2018-10-31
> **Word Vocabulary**
>
> Hi Lucas,
>
> Thank you for the comment and the interesting reference. In our Librispeech experiments, OCD significantly outperforms MLE when using a vocabulary of 10K tokens generated using Byte Pair Encoding (BPE). Many of these tokens are complete words. We expect similar gains over the MLE baseline when using a vocabulary of words. However, BPE has received more attention than word based encoding in the NLP community recently because of its seamless handling of unknown tokens.
>
> We note that OCD can optimize word edit distance directly on a character-based model as well, by modifying the proposed dynamic programming algorithm. That being said, since OCD performs well on BPE, we think direct optimization of token edit distance based on BPE tokens is well suited for many real world applications.

---

> > ### Public Comment · (anonymous) · 2018-11-03
> > **Thanking the authors**
> >
> > Thank you for the quick answer! Hope your paper gets in.
> > -Lucas

---

### Public Comment · ~Murali_Karthick_Baskar1 · 2018-11-07
**Nicely suited fro ASR**

Thanks for a nice paper. After reading your paper, I found it might be closer to the previous work in ASR in the following paper.
Link:  http://www.mirlab.org/conference_papers/international_conference/ICASSP%202018/pdfs/0005839.pdf

---

> ### Author Response · Authors · 2018-11-24
> **Thanks, updated Related work accordingly.**
>
> Thank you for the pointer. We have added a citation and included a discussion in the new version of the paper.

---

### Author Response · Authors · 2018-11-24
**Revision updates**

We are deeply grateful for the feedback we received. The paper has been revised to address all of the comments. The newly added parts of the paper are highlighted in green to enable easier comparison with the previous version. Our main updates are as follows:

1) Improved and clarified the exposition of the objective function around Equation (6).
2) Expanded and clarified the Related Work section and added the missing citations.
3) Added the training & validation WER curves for MLE and OCD as Figure 3 to verify that even early on, OCD has stable training.

We are happy that the paper is well received by the research community so far.
-- Authors

---

### Public Comment · ~Saeed_Najafi1 · 2019-06-03
**code?**

This is an awesome unifying approach between RL, EditDistance, and Imitation Learning.
Will authors release the code as well?
Meanwhile, I've been trying to implement the core part of the algorithm at https://github.com/SaeedNajafi/pytorch-ocd
-- Thank You

---

### Meta-Review · Area_Chair1 · 2018-12-13
**Exciting approach to training sequence-to-sequence models from scratch**

**Confidence:** 5
**Recommendation:** Accept (Poster)

**Metareview:**

This paper proposes an algorithm for training sequence-to-sequence models from scratch to optimize edit distance. The algorithm, called optimal completion distillation (OCD), avoids the exposure bias problem inherent in maximum likelihood estimation training, is efficient and easily implemented, and does not have any tunable hyperparameters. Experiments on Librispeech and Wall Street Journal show that OCD improves test performance over both maximum likelihood and scheduled sampling, yielding state-of-the-art results. The primary concerns expressed by the reviewers pertained to the relationship of OCD to methods such as SEARN, DAgger, AggreVaTe, LOLS, and several other papers. The revision addresses the problem with a substantially larger number of references and discussion relating OCD to the previous work. Some issues of clarity were also well addressed by the revision.